# ON THE GENERALIZATION EFFECTS OF DENSENET MODEL STRUCTURES

## ABSTRACT

Modern neural network architectures take advantage of increasingly deeper layers, and various advances in their structure to achieve better performance. While traditional explicit regularization techniques like dropout, weight decay, and data augmentation are still being used in these new models, little about the regularization and generalization effects of these new structures have been studied. Besides being deeper than their predecessors, could newer architectures like ResNet and DenseNet also benefit from their structures' implicit regularization properties? In this work, we investigate the skip connection's effect on network's generalization features. Through experiments, we show that certain neural network architectures contribute to their generalization abilities. Specifically, we study the effect that low-level features have on generalization performance when they are introduced to deeper layers in DenseNet, ResNet as well as networks with 'skip connections'. We show that these low-level representations do help with generalization in multiple settings when both the quality and quantity of training data is decreased.

## 1 INTRODUCTION

Deep models have achieved significant success in many applications. However, deep models are hard to train and require longer times to converge. A solution by construction is copying the learned layers from the shallower model and setting additional layers to identity mapping. Skip connection proposed in the Residual Network He et al. (2016), shows the new insight of innovation in network structure for computer vision.

In the following years, more new and multi-layer-skipping structures have been proposed and proved to have better performance, among which one typical example is DenseNet (Huang et al., 2016). ResNet (He et al., 2016), HighwayNet (Rupesh Kumar Srivastava & Schmidhuber, 2015) and FractalNets (Larsson et al., 2016) have all succeeded by passing the deep information directly to the shallow layers via shortcut connection. Densenet further maximize the benefit of shortcut connections to the extreme. In DenseNet (more accurately in one dense block) every two layers has been linked, making each layer be able to use the information from all its previous layers. In doing this, DenseNet is able to effectively mitigate the problem of gradient vanishing or degradation, making the input features of each layer various and diverse and the calculation more efficient.

Concatenation in Dense Block: the output of each layer will concatenate with its own input and then being passed forward to the next layer together. This makes the input characteristics of the next layer diversified and effectively improves the computation and helps the network to integrate shallow layer features to learn discriminative feature. Meanwhile, the neurons in the same Dense block are interconnected to achieve the effect of feature reused. This is why DenseNet does not need to be very wide and can achieve very good results.

Therefore, shortcut connections form the multi-channel model, making the flow of information from input to output unimpeded. Gradient information can also be fed backward directly from the loss function to the the various nodes.

In this paper we make the following contributions:

- We design experiments to illustrate that on many occasions it is worth adding some skip connections while sacrificing some of the network width. Every single skip connection replacing some of width is able to benefit the whole network's learning ability. Our 'connection-by-connection' adding experiment results can indicate this well.

- We perform experiments to show that networks that reuse low-level features in subsequent layers perform better than a simple feed-forward model. We degrade both the quantity and the quality of the training data in different settings and compare the validation performances of these models. Our results suggest that while all models are able to achieve perfect training accuracy, both DenseNet and ResNet are able to exhibit better generalization performance given similar model complexities.

- We investigate solutions learned by the three types of networks in both a regression and classification involving task in low dimensions and compare the effects of both the dense connections and the skip connections. We show that the contribution of the feature maps reintroduced to deeper layers via the connections allow for more representational power.

## 2    THE EFFECTS OF SKIP CONNECTIONS

Skip connections are the main features of DenseNet and ResNet, which is convenient to make gradient flow easily and overcome the overfitting. For this reason, it is interesting and necessary to dive more in the effects of skip connections on the performance.

In the original DenseNet paper (Huang et al., 2016), three dense blocks are connected sequentially and in each dense block all of the convolutional layers have direct paths connected with each other.

In our implementation, there are 8 layers which leads to a total of 28 skip connections within each dense block. We increase the number of skip connections from 0 to 28 and test the validation accuracy on CIFAR100 trained with 10k and 50k samples. The total numbers of parameters in these models are set to the same (92k) by controlling the depths of convolutional filters in each dense block.

Figure 1 shows the variation of validation accuracy with changing number of skip connections. Despite some fluctuations in numerical values, the overall trend of validation accuracy goes up, showing the increasing number of skip connections have positive effects on the generalization performance of neural networks.

Experiments are conducted to test the effect of skip connections in the next section. We will change the number of skip connections in network smoothly.

The increasing process of the number of skip connections can be described as follows: For each layer, first of all the connections to their previous layers which is 2 layers away is added. Then the connections linking layers farther away from each other is added.

## 3    EXPERIMENTS

### 3.1    GENERALIZATION PERFORMANCE IN TROUBLESOME DATASET

In this section the generalization performances of three different network structures, Cascade Network(simple layer-wise CNN), ResNet and DenseNet, using MNIST and CIFAR100 dataset are measured. The experiments are done by modifying these datasets either by decreasing the number of training samples or by adding noise to groundtruth labels so as to test which network structure shows better performance under these 'harsh training conditions'.

#### 3.1.1    TRAINING WITH DECREASED SAMPLES

Neural networks is prone to overfitting if the training data fed into the network is insufficient. In order to compare generalization performances of the three networks with different connectivity patterns, the appropriate depth and parameters for these networks are selected carefully for better comparisons.

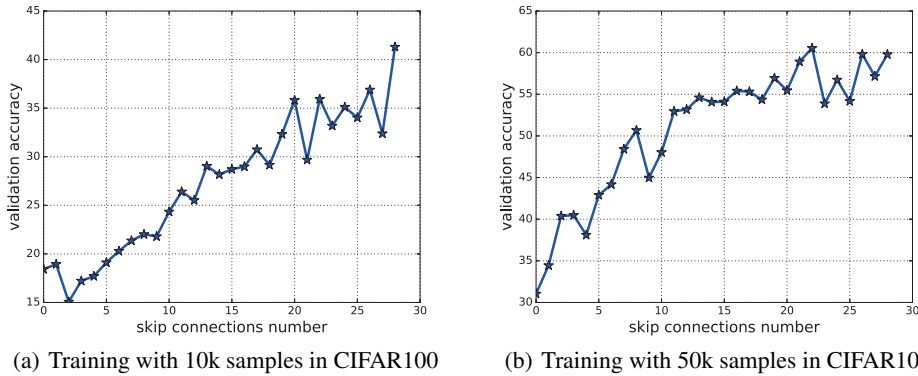

(a) Training with 10k samples in CIFAR100  (b) Training with 50k samples in CIFAR100

Figure 1: Validation accuracy with different number of skip connections in CIFAR 100.

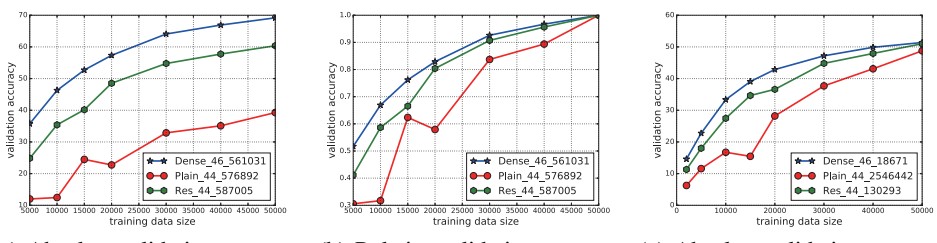

(a) Absolute validation accuracy  (b) Relative validation accuracy  (c) Absolute validation accuracy

Figure 2: Validation accuracy with different network architectures in CIFAR100. (a) Absolute validation accuracy with same number of parameters. (b) Relative validation accuracy with same number of parameters. (c) Absolute validation accuracy with different number of parameters.

The depths of the networks range from 44 to 46 layers due to the fact that it has a fair learning capacity. We haven't chosen the standard DenseNet or ResNet with over a hundred layers as described in the original papers because the plain Cascade Network will suffer from severer gradient vanishing problems as the number of layers increases. The numbers of channels of each layer are carefully picked so that sizes of the three model's parameters are nearly the same (560k to 580k in our experiment) to ensure fairness of our comparisons.

Figure 2(a) shows the absolute validation accuracy of the three networks with varying sizes of training data. It can be seen that with a large number of skip connections, DenseNet outperforms other network structures on all sizes of training data. The three curves are then normalized in Figure 2(b) for better visual comparison, which is done by scaling each line's maximum value in Figure 2(a) to 100%.

In Figure 2(c), the width of each network is specially designed so that when the entire training set is used for training the three networks, similar performance could be reached, which is all around 50%. From the figure it can be noticed that while Cascade Net (Plain Net) has much more parameters, it is the least robust to the decreasing number of training samples. On the contrary, the DenseNet, with more skip connections, has the fewest parameters but the highest robustness.

With the decrease number of training samples, the performance of DenseNet drops more moderately than the others shown in Figure 2. The architecture can benefit the network's stability and its generalization. As a consequence, the dense connection can be a good method of generalization.

The similar results have been achieved in MNIST, where the only difference is that the networks contains only fully connected layers without any convolution layers. Considering the potential over fitting problem of deep Linear network and the simplicity of the dataset, we construct just 6 layers for Cascade Net and DenseNet. Same as CIFAR dataset, we vary the widths in order to achieve

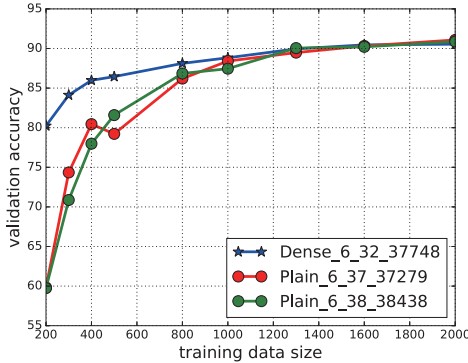

Figure 3: The absolute validation accuracy with varying training data size of different network structures in MNIST dataset

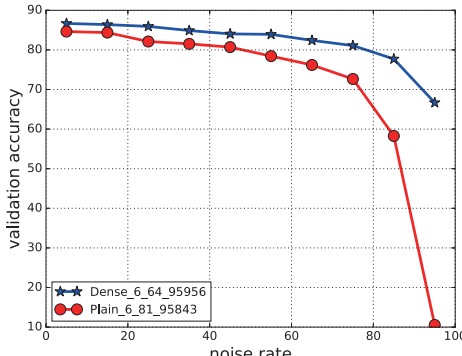

Figure 4: The validation accuracy with varying noise rate of different network structures in MNIST dataset

similar numbers of parameters among the three network architectures. Figure 3 illustrates that the dense skip connections have made the network more robust against the paucity of the training data size.

We can conclude from the above experiments that for deep neural networks both with and without convolution layers, the skip connections can effectively act as an implicit regularizer in network training process, making networks generalize better.

### 3.1.2 TRAINING WITH NOISE IN DATASET

To test the skip connections' 'adapting skills', we have tried to add some noise to training dataset. The noise is added by directly setting some pixels in some channels as 0. The result is as the noise grows bigger, the decrease in DenseNet's performance will be smaller than others. As shown in Figure 4, network with dense connections is more robust to noise.

### 3.2 VISUALIZATION OF GENERALIZATION EFFECTS IN ONE DIMENSION

In this section we visualize the generalization effects of network structures using the simple 1-dimensional curve fitting problem. The network has to learn a function $f$ that maps scalar value $x$ to scalar value $f(x)$. Network with better regularization capability should have smoother output in the hidden layers and may thus avoid the over-fitting problem, producing greater generalization ability.

We use the simple networks which contains only fully-connected layer and the nonlinear layers to fit the Sinc Function defined as: $f(x) = \frac{\sin x}{x}$ in range of $(-15, 15)$. Three kinds of networks

are implemented using their main characteristics: the Cascade Net uses a smooth linear model, the ResNet in which several skip connection is added, and the DenseNet which has only one dense block and between every two layers there is one direct connection.

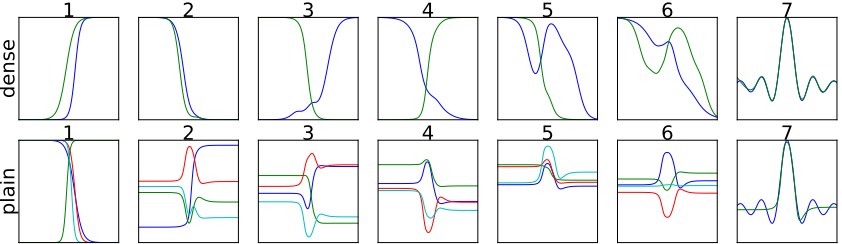

Figure 5: Learning process of sinc curve (training data number for DenseNet is 30 while for Plain Net is 400)

In order to analyze the results more in detail, we both plot the network's final learned curve and extract the output from each layer.

The parameters of all 3 kinds of networks of the same depth are controlled as near as possible, by adjusting the 'width' ('growth rate'). For example, when growth rate is 2 for DenseNet and 4 for Plain Net, their parameters are 98 and 113 respectively. Though the Plain Net has more parameters, its results are unsatisfactory.

*Top sequence* of subplots in Figure 5 are the learning process of 7-layer dense net with a training sample of 30 points. Each subplot is the output of each layer (as the growth rate is 2 there are 2 curves in each plot). Also, while there is only 30 training data points, it can still learn well. In the $7^{th}$ subplot, the blue curve is the standard sinc curve and the green one is its learned version where it can be seen that the two are almost identical. The last plot is the training loss changing with the epochs.

As for the ordinary Plain Net, the result with 30 training data points or even 200 training data points is always very bad without fitting the small waves of both sides.

The output 4 curves from each layer is also as simple but the final results are bad. It couldn't learn the trivial wave. The thing that is also not satisfactory is in its training loss which drops quickly at the beginning but cannot be smaller later. When more training data are fed to the Plain Net, *sometimes* it can do a good job. However, most times it still cannot learn more information, like in *bottom sequence* of subplots in Figure 5 with 400 training data, it is still quite hard for this Plain Net to compete with the dense one.

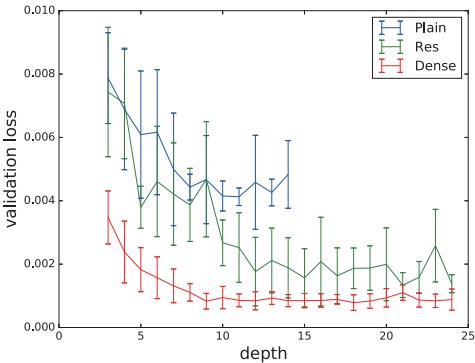

Figure 6: Loss with depth—statistical results of noisy training data (training data number: 60).

Then some advancements are made:

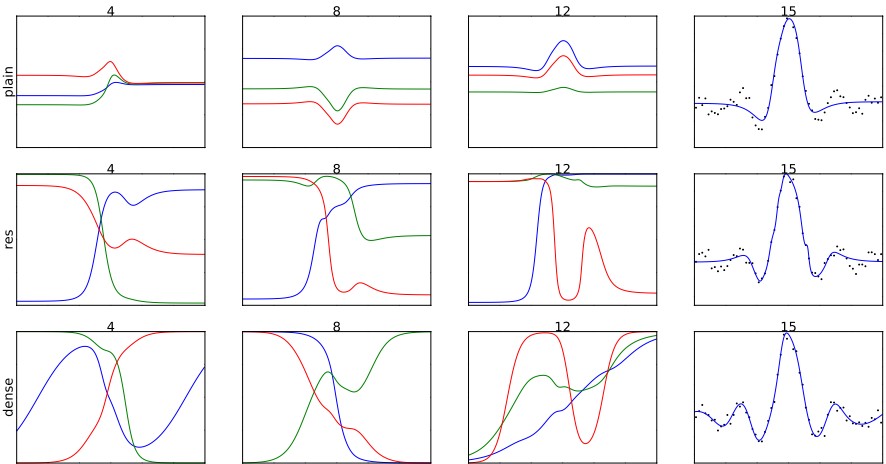

Figure 7: 15-layer Nets' learning process of sinc curve (noise training data number: 60).

- The ResNet is added to them to compare together. At least residual net is of most competitiveness to DenseNet.

- Some noise is added to the training data (the training data points are with a small fluctuation from its original sinc curve). With the noise, it is more interesting to see the networks' generalization. The results are similar as the circumstance with no noise.

- Same experiments are conducted many a time which enables us to plot some statistical results. In Figure 6, x-axis is the depth and y-axis is the final loss. The middle point is the mean loss in 10 experiments, the line segments represent the deviation in these same experiments.

Still, in these experiments it is designed to make sure that in each depth the parameter number of two kinds of nets are similar through adjusting the growth rate. On one hand, it can be seen that DenseNet is better when deeper. On the other, the Plain Net learns nothing when it is too deep. It is due to the gradient vanishing as will be discussed later.

No matter in the mean loss or the deviation, the dense net is smaller than the other two. And It can generalize even much better than residual net relatively when the nets are shallow.

We can have a closer look at 15-layer nets of each type in Figure 7. In order to make the parameters numbers of 3 types nearly identical to each other, the Plain Net and the ResNet each has 8 lines of output of previous layer while the DenseNet has only 3 lines (growth rate of each layer). In the figure, only 3 of them are chosen to be displayed to have a more equitable comparison with DenseNet. Instead of showing every layer's output we exhibit the $3^{th}$, $5^{th}$, $7^{th}$, $9^{th}$, $11^{th}$, $13^{th}$ and the final layer's output. Also, the training data plotted on the final output is 60 points with trivial deviation from the standard sinc curve, together with the network's learnt curve.

- Plain Net:
  The worst one as can be seen on the *top* row in Figure 7: gradient vanishing is severe and finally it learns just a little profile of the curve which makes no sense at all.

- Net with residual connection:
  This is much better than the Plain Net (the *middle* row of Figure 7. It can fit the curve well in subplot 20. However, the middle output seems a bit complicated and its final fitting is not perfect either.

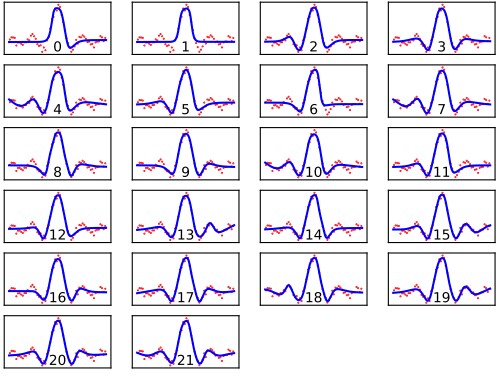

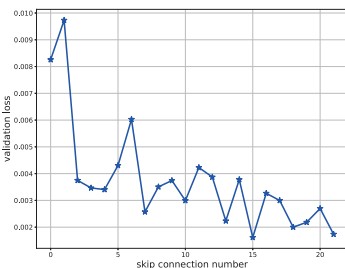

(a) Curve Fitting, transition from Plain Net to Most Dense Net

(b) Validation Loss of the networks with different density

Figure 8: Validation accuracy with different network architectures in CIFAR100. (a) Absolute validation accuracy with same number of parameters. (b) Relative validation accuracy with same number of parameters. (c) Absolute validation accuracy with different number of parameters.

- Net with dense connection:
  To make number of parameters similar to the two nets above, its growth rate is only 3. It can also fit the curve even better and the output of previous layers is the simplest as the *bottom* row of Figure 7.

For better investigate the effects of 'skip connection', a more detailed experiment has been conducted.

In previous experiments, we only compare no more than 3 kinds of different network architectures, which may not be concrete enough. Thus, we exploit more forms of networks by split the numbers of skip connections. Their 'density' is between the two extreme. The skip connections are added one by one, where each defines a different network architecture. Network widths are chosen accordingly to make the numbers of parameters not differ much. The depth is 7 for all architectures, which means there will be at most 21 'skip connections'. As can be seen in Figure 8(a), the fitting effect is becoming better as the network becomes 'denser'. The corresponding loss is displayed in Figure 8(b). From these two figures, the representational power of skip connections is conspicuous.

## 3.3 VISUALIZATION OF GENERALIZATION EFFECTS IN TWO DIMENSIONS

We move on to analyzing the three styles of networks above in a two-dimensional classification problem. The networks have to learn a decision boundary from a non-linearly separable data set. We again restrict our depth to eight layers with the number parameters across the Dense, Residual, and the Plain networks being 614, 712, and 712, respectively. The parameters for the Dense network is controlled by adjusting the growth rate of each layer. Using the same hyper-parameters and number of training epochs, we attain the intermediate decision boundaries across each layer of each network to see the progression of complexity with increasing depth. Similar to the one-dimensional experiments, networks that generalize better tend to learn smooth decision boundaries in the presence of noise rather than over-fitting to all data points.

To visualize the intermediate results of the networks, we feed a grid of evenly separated data points in the vicinity of our original two-dimensional data points and record the raw outputs after being activated by the activation function in each of the layers. Since only the last layer, i.e., the output layer, has two-dimensional outputs, we choose one of the dimensions in each layer to visualize. The top row in Figure 9 shows the progression of a densely connected network in the style of

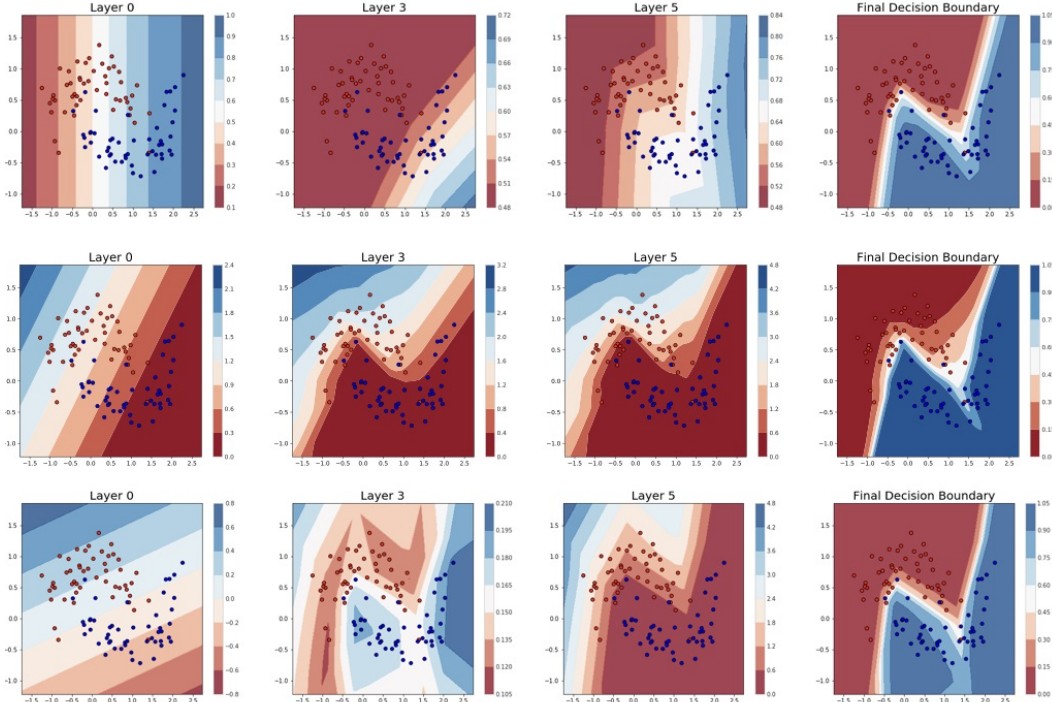

Figure 9: Intermediate decision boundaries for three different models.

DenseNet(Huang et al., 2016). Notice that in the Dense network, which achieves the lowest lost in the test set, every layer receives inputs from all preceding layers, and is, therefore, able to make use of low-level features even at the last layer stage. The first row of the figure shows the intermediate features received by the eighth layer, which includes the linear features like those from the first layer, all the way to higher-level features from the third and the fifth. We decide to show this last layer for this network since it encompasses learning from all previous stages.

The benefits of dense connections, however, is not present in the Residual and the Plain networks. The second and third rows of Figure 9 show the features learned in the first, third, and the fifth layers.

## 4  CONCLUSION

By introducing skip connections, modern neural network has proved better performance in computer vision area. This paper investigates how skip connections works in vision task and how they effect the learning power of networks. For this reason, we have design some experiments and verify that networks with skip connections can do the regression best among tested network architectures. It indicates that we can get the insights of this interesting architecture and its tremendous learning power.

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
