# OpenReview forum: "On the Generalization Effects of DenseNet Model Structures "
_ICLR.cc/2018/Conference — Reject_

### Official Review · AnonReviewer2 · 2017-11-27
**analysing skip connections**

**Rating:** 2
**Confidence:** 5

**Review:**

The ms analyses a number of simulations how skip connections effect the generalization of different network architectures. The experiments are somewhat interesting but they appear rather preliminary. To indeed show the claims made, error bars in the graphs would be necessary as well will more careful and more generic analysis. In addition clear hypotheses should be stated.
The fact that some behaviour is seen in MNIST or CIFAR in the simulations does not permit conclusion for other data sets. Typically extensive teacher student simulations are required to validly make points. Also formally the paper is not in good shape.

---

### Official Review · AnonReviewer1 · 2017-11-27
**Analysis paper on the generalization effect of skip connections, motivation and contributions are not clear, references are very limited**

**Rating:** 3
**Confidence:** 4

**Review:**

This paper analyzes the role of skip connections with respect to generalization in recent architectures such as ResNets or DenseNets. The authors perform an analysis of the performance of ResNets and DenseNets under data scarcity constraints and noisy training samples. They also run some experiments assessing the importance of the number of skip connections in such networks.

The presentation of the paper could be significantly improved. The motivation is difficult to grasp and the contributions do not seem compelling.

My main concern is about the contribution of the paper. The hypothesis that skip connections ease the training and improve the generalization has already been highlighted in the ResNet and DenseNet paper, see e.g. [a].

[a] https://arxiv.org/pdf/1603.05027.pdf

Moreover, the literature review is very limited. Although there is a vast existing literature on ResNets, DenseNets and, more generally, skip connections, the paper only references 4 papers. Many relevant papers could be referenced in the introduction as examples of successes in computer vision tasks,  identity mapping initialization, recent interpretations of ResNets/DensetNets, etc.

The title suggests that the analysis is performed on DenseNet architectures, but experiments focus on comparing both ResNets and DenseNets to sequential convolutional networks and assessing the importance of skip connections.

In section 3.1. (1st paragraph) proposes adding noise to groundtruth labels; however, in section 3.1.2,. it would seem that noise is added by changing the input images (by setting some pixel channels to 0). Could the authors clarify that? Wouldn’t the noise added to the groundtruth act as a regularizer?

In section 4, the paper claims to investigate the role of skip connections in vision tasks. However, experiments are performed on MNIST, CIFAR100, a curve fitting problem and a presumably synthetic 2D classification problem. Performing the analysis on computer vision datasets such as ImageNet would be more compelling to back the statement in section 4.

---

### Official Review · AnonReviewer3 · 2017-11-27
**An analysis paper with only 4 citations.**

**Rating:** 3
**Confidence:** 4

**Review:**

The paper studies the effect of different network structures (plain CNN, ResNet and DenseNet). This is an interesting line of research to pursue, however, it gives an impression that a large amount of recent work in this direction has not been considered by the authors. The paper contains ONLY 4 references.

Some references that might be useful to consider in the paper:
- K. Greff et. al. Highway and Residual Networks learn Unrolled Iterative Estimation.
- C. Zang et. al. UNDERSTANDING DEEP LEARNING REQUIRES RETHINKING GENERALIZATION
- Q. Liao el. al. Bridging the Gaps Between Residual Learning, Recurrent Neural Networks and Visual Cortex
- A. Veit et. al. Residual Networks Behave Like Ensembles of Relatively Shallow Networks
- K. He at. Al Identity Mappings in Deep Residual Networks

The writing and the structure of the paper could be significantly improved. From the paper, it is difficult to understand the contributions. From the ones listed in Section 1, it seems that most of the contributions were shown in the original ResNet and DenseNet papers. Given, questionable contribution and a lack of relevant citations, it is difficult to recommend for acceptance of the paper.

Other issues:
Section 2: “Skip connection …. overcome the overfitting”, could the authors comment on this a bit more or point to relevant citation?
Section 2: “We increase the number of skip connections from 0 to 28”, it is not clear to me how this is done.
Section 3.1.1 “deep Linear model”, what the authors mean with this? Multiple layers without a nonlinearity? Is it the same as Cascade Net?
Section 3.2 From the data description, it is not clear how the training data was obtained. Could the authors provide more details on this?
Section 3.2 “…, only 3 of them are chosen to be displayed…”, how the selection process was done?
Section 3.2 “Instead of showing every layer’s output we exhibit the 3th, 5th, 7th, 9th, 11th, 13th and the final layer’s output”, according to the description in Fig. 7 we should be able to see 7 columns, this description does not correspond to Fig. 7.
Section 4 “This paper investigates how skip connections works in vision tasks…” I do not find experiments with vision datasets in the paper. In order to claim this, I would encourage the authors to run tests on a CV benchmark dataset (e. g. ImageNet)

---

### Decision · Program_Chairs · 2018-01-29
**ICLR 2018 Conference Acceptance Decision**

**Decision:**

Reject

**Comment:**

The paper appears unfinished in many ways: the experiments are preliminary, the paper completely ignored a large body of prior work on the subject, and the presentation needs substantial improvements. The authors did not provide a rebuttal.

I encourage the authors to refrain from submitting unfinished papers such as this one in the future, as it unnecessarily increases the load on a review system that is already strained.